# Characterization of BDS Multipath Effect Based on AT-Conv-LSTM Network

**Jie Sun, Zuping Tang, Chuang Zhou and Jiaolong Wei \***

School of Electronic Information and Communication, Huazhong University of Science and Technology, Wuhan 430074, China; jessiesun@hust.edu.cn (J.S.); tang_zuping@hust.edu.cn (Z.T.); zhouchuang@hust.edu.cn (C.Z.)
**\*** Correspondence: jlwei@hust.edu.cn

**Abstract:** Multipath effects are the most challenging error sources for the Global Navigation Satellite System receiver, affecting observation quality and positioning accuracy. Due to the non-linear and time-varying nature, multipath error is difficult to process. Previous studies used a homogeneous indicator to characterize multipath effects and only revealed the temporal or spatial correlations of the multipath, resulting in limited correction performance. In this study, we consider the code multipath to be influenced not only by the elevation and azimuth angle of certain stations to satellites but also to be related to satellite characteristics such as nadir angle. Hence, azimuth angle, elevation angle, nadir angle and carrier-to-noise power density ratio are taken as multiple indicators to characterize the multipath significantly. Then, we propose an Attention-based Convolutional Long Short-Term Memory (AT-Conv-LSTM) that fully exploits the spatiotemporal correlations of multipath derived from multiple indicators. The main processing procedures using AT-Conv-LSTM are given. Finally, the AT-Conv-LSTM is applied to a station for 16 consecutive days to verify the multipath mitigation effectiveness. Compared with sidereal filtering, multipath hemispherical map (MHM) and trend-surface analysis-based MHM, the experimental results show that using AT-Conv-LSTM can decrease the root mean square error and mean absolute error values of the multipath error more than 60% and 13%, respectively. The proposed method can correct the code multipath to centimeter level, which is one order of magnitude lower than the uncorrected code multipath. Therefore, the proposed AT-Conv-LSTM network could be used as a powerful alternative tool to realize multipath reduction and will be of wide practical value in the fields of standard and high-precision positioning services.

**Keywords:** BDS; multipath; AT-Conv-LSTM; spatiotemporal domain





## 1. Introduction

As an essential technology, the Global Navigation Satellite System (GNSS) has been utilized in various fields, including aviation [1], geodesy [2] and earthquake detection [3] and climate monitoring [4]. However, in complex environments like urban canyons, GNSS stations face challenges due to the reflection, diffraction and obstruction of signals by local obstacles around the antenna [5]. Currently multipath does not have a commonly accepted method for its correction that introduces meter-scale code observation errors and centimeter-scale carrier phase observation errors [6]. Hence, it is crucial to develop a method for mitigating multipaths to enhance the accuracy of standard and high-precision positioning services.

For certain environments and receivers, there are two distinct multipath mitigation classifications: hardware enhancement and data handling. The former methods principally include antenna-based designs [7,8] and receiver-based architectures [9] to mitigate multipath errors. However, the enhancement has limited effectiveness and are difficult to implement. The data handling methods eliminated or mitigated multipath errors via code and phase observations combination [10,11], parameterization [12] or model correction.

The parameterization category chooses the carrier-to-noise power density ratio (C/N0) and signal-to-noise ratio (SNR) [13,14], parameterized into the stochastic model to characterize multipath effects.

One kind of empirical model correction is based on time-domain repeatability of multipath, such as sidereal filtering (SF) [15]. Genrich and Bock proposed the SF method and calculated the orbital repeat time (ORT) of GPS satellites to be 23 h, 56 min and 4 s to establish multipath correction model. Choi et al. regarded the ORT should be calculated for individual satellite [16], which revealed that the sidereal repeat period deviates from the widely accepted sidereal recurrence by roughly 9 s. The position-domain SF method has been introduced to attain multipath mitigation in precise point positioning (PPP) [17]. Ragheb et al. carried out position-domain and observation-domain multipath mitigation methods for GPS and applied them to the precise point position model [18]. Hung and Rau improved the multipath corrected efficiency via bandpass filters in the position domain [19]. In addition, Atkins and Ziebart evaluated and compared the effectiveness of observation domain SF with position domain SF for GPS PPP. Different from the position domain, these variations of orbital repetition time are slightly different for individual GPS satellites in the observation domain. And observation-domain multipath mitigation is more advantageous in the handling of high-frequency components [20]. However, the sidereal filtering requires the pre-computation of the orbital repetition time, thereby deteriorating the real-time performance and positioning accuracy of multipath mitigation.

Another approach used in the literatures is based on spatial-domain repeatability. Considering that the position of GNSS antenna and the environment around the GNSS stations remain unchanged, the multipath is solely in accordance with the particular elevation and azimuth angle of station-satellite couple. The emerging methods consist of multipath spherical harmonic model [21], multipath stacked (MPS) model [22] and multipath hemispherical diagram (MHM) [23]. The multipath correction value for the MPS and MHM approaches is the average of all satellite residuals in each grid. These approaches realize $1° \times 1°$ high-resolution equal lattice grids on the sky map, which are easier to produce than for the spherical harmonic model. Multipath hemispherical diagram based on trend-surface (T-MHM) is proposed to fit the multipath spatial distribution with trend-surface modeling within the grid [24]. Lu et al. discussed the optimal modeling days, applied T-MHM to BDS-3 PPP and evaluated the multipath correction effect simultaneously, and validated the mitigation effect of T-MHM on different grid scales [25,26]. Zhang et al. evaluated the multipath mitigation effect of MHM in BDS2/BDS3 real-time kinematic (RTK) [27]. Although this method has a low complexity, it disregards the multipath's spatial distribution within the grid. Thus, it is more effective in correcting low frequency multipath but has limited effectiveness in correcting the high-frequency multipath. It is clear that averaging residuals could somewhat filter out high-frequency signals in the grids. This drawback can be resolved by reducing the grid but introduces the new risk of rendering the model less robust.

The majority of research and studies on BDS-3 multipath mitigation have mainly been conducted on the different observation model, whereas few have explored BDS-3 multipath mitigation strategy on a undifferenced and uncombined PPP model. When existing methods are used to mitigate the BDS with hybrid constellation, the multipath model will be more complex than GPS systems. In current studies, SNR or C/N0 are used to evaluate the observations and identify the multipath, elevation and azimuth angle, which are other indicators used to the characterize multipath in the MHM method. However, the indicator to reflect the characteristics of multipath effects in current studies is comparatively homogeneous, the multiple joint indicators are critical to guaranteeing the best performance. In this study, we use the azimuth angle, elevation angle, nadir angle and C/N0 as multiple indicators to characterize the multipath significance in undifferenced and uncombined PPP modes.

The modeling and mitigation of the multipath pose significant challenges due to its complex nonlinear and time-varying nature. In recent years, deep learning has emerged

as a powerful technique for addressing non-linear problems and has been successfully employed in various domains, such as ionosphere forecasting [28,29], troposphere tomography [30], satellite orbit broadcast [31], satellite clock prediction [32], self-driving [33] and integrated navigation [34]. Deep learning algorithms such as neural networks are data-driven models that use large and extensive datasets to obtain correlations without relying on complex physically based models [35]. Moreover, multiple indicators make it more difficult to establish the multipath mathematical model. The attention mechanism is able to focus on the important information from the massive datasets and ignore mostly dispensable information [36,37]. The attention mechanism can help the network in assigning weights of the TEC time series to forecast ionospheric TEC [38]. The attention-based Conv-LSTM network is proposed, CNN and LSTM modules are applied to obtain the spatial feature and the temporal feature, respectively, while the attention mechanism could emphasize importance level in the dataset [39]. Recently, few researchers have utilized artificial intelligence in the multipath mitigation field [35]. But existing methods only reveal the temporal or spatial correlations of multipath, which fail to fully grasp the elaborate characteristics of individual time series and has limited enhancement in mitigation performance. Therefore, we propose a multipath mitigation using the Attention-based Convolutional Long Short-Term Memory (AT-Conv-LSTM) to maximize the spatiotemporal repeatability of multipath with multiple indicators.

In this study, the potential of utilizing AT-Conv-LSTM for mitigating multipath effects is elaborated. Firstly, we derive the multipath extracting method on the undifferenced and uncombined PPP model and analyze multiple indicators, including elevation, azimuth, nadir angle and C/N0 to characterize the multipath spatiotemporal correlation. Secondly, we propose the multipath mitigation method based on AT-Conv-LSTM network, and the main pre-processing steps are outlined. Then, the comparative experiment is used to analyze the multipath discrepancy of SF, MHM, T-MHM and AT-Conv-LSTM. Finally, the study concludes with a summary and suggestions for future research.

## 2. Multipath Analysis Method

### 2.1. Multipath Extraction

The code and carrier observations of the GNSS signal received at the station are:

$$p_{r,i}^{s,Q} = \mathbf{u}_r^{s,Q} \cdot \mathbf{R}_r^s + dt_r^s - dt^{s,Q} + M_{\mathrm{w}} \cdot Z_{\mathrm{w}} + \gamma_i^Q \cdot I_1^{s,Q} + b_{r,i}^s - b_i^{s,Q} + m_{r,i}^{s,Q} + \varepsilon_{r,i}^{s,Q}, \quad (1)$$

$$l_{r,i}^{s,Q} = \mathbf{u}_r^{s,Q} \cdot \mathbf{R}_r^s + dt_r^s - dt^{s,Q} + M_{\mathrm{w}} \cdot Z_{\mathrm{w}} - \gamma_i^Q \cdot I_1^{s,Q} + \lambda_i^s \cdot N_i^{s,Q} + B_{r,i}^s - B_i^{s,Q} + M_{r,i}^{s,Q} + \xi_{r,i}^{s,Q}, \quad (2)$$

where $p_{r,i}^{s,Q}$ and $l_{r,i}^{s,Q}$ indicate the values of "observed minus computed (OMC)" for code and carrier phase observables, individually; $s$ represents the PRN number, $Q$ represent the satellite system, $r$ expresses the receiver ID, $i$ expresses the frequency band number; $\mathbf{u}_r^{s,Q}$ denotes the line-of-sight (LOS) unit vector; $\mathbf{R}_r^s$ is the vector of receiver position increments relative to the a priori position; $dt_r^s$ states the receiver clock offsets, $dt^{s,Q}$ states the satellite clock offsets; $Z_{\mathrm{w}}$ means the zenith wet delay; $M_{\mathrm{w}}$ means the wet mapping function; $I_1^{s,Q}$ implies the ionospheric delay on the first frequency band, $\gamma_i^Q = \left(\frac{f_1^{s,Q}}{f_{iW}^{s,Q}}\right)^2$ is the multiplier factor introduced to convert to frequency $f_i$; $b_{r,i}^s$ indicates the receiver uncalibrated code delay (UCDs) and $b_i^{s,Q}$ indicates the satellite UCDs corresponding to frequency $f_i$; $\lambda_i^s$ denotes the carrier wavelength on $i$th frequency band; $N_i^{s,Q}$ denotes the integer phase ambiguity; $B_{r,i}^s$ and $B_i^{s,Q}$ are the receiver and satellite uncalibrated phase delays (UPDs), which is different on each frequency band; the code and phase multipath error can be expressed as $m_{r,i}^{s,Q}$ and $M_{r,i}^{s,Q}$; and $\varepsilon_{r,i}^{s,Q}$ and $\xi_{r,i}^{s,Q}$ represent the code and carrier measurement noise. The receiver and satellite antenna phase center offsets (PCOs) and variations (PCVs), phase windup, tidal loadings, LOS hydrostatic delay, relativistic effects and Sagnac effects should be corrected via empirical models.

For convenience, the coefficient for the ionosphere-free (IF) combination are defined as:

$$\alpha_{mn}^Q = \frac{\left(f_m^{s,Q}\right)^2}{\left(f_m^{s,Q}\right)^2 - \left(f_n^{s,Q}\right)^2}$$

$$\beta_{mn}^Q = -\frac{\left(f_n^{s,Q}\right)^2}{\left(f_m^{s,Q}\right)^2 - \left(f_n^{s,Q}\right)^2} \tag{3}$$

where $f^{s,Q}$ stands for the frequency band ($m$, $n = 1, 2; m \neq n$) and $\alpha_{nm}^Q$ and $\beta_{nm}^Q$ are the IF combination coefficients.

When linearly coupled with the satellite UCD, the satellite clock offset cannot be separated independently unless excess baseline constraints are incorporated. Currently, the IGS precise satellite clock products are estimated by utilizing the IF combination observables. Consequently, the satellite clock offsets containing the IF combination of satellite UCDs are:

$$\begin{aligned} dt^s_{\mathrm{IF}_{12}^{s,Q}} &= dt^{s,Q} + \left(\alpha_{12}^Q \cdot b_1^{s,T} + \beta_{12}^Q \cdot b_2^{s,Q}\right) \\ &= dt^{s,Q} + b_{\mathrm{IF}_{12}}^{s,Q}, \end{aligned} \tag{4}$$

with

$$\begin{cases} DCB_{P_mP_n}^{s,Q} = b_m^{s,Q} - b_n^{s,Q}, DCB_{r,P_mP_n}^{s,Q} = b_{r,m}^{s,Q} - b_{r,n}^{s,Q} \\ b_{IF_{mn}}^{s,Q} = \alpha_{mn}^Q \cdot b_m^{s,Q} + \beta_{mn}^Q \cdot b_n^{s,Q}, b_{r,IF_{mn}}^{s,Q} = \alpha_{mn}^Q \cdot b_{r,m}^{s,Q} + \beta_{mn}^Q \cdot b_{r,n}^{s,Q} \end{cases}, \tag{5}$$

where $DCB_{P_mP_n}^{s,Q}$ and $DCB_{r,P_mP_n}^{s,Q}$ express satellite and receiver DCB between pseudoranges $p_{r,m}^{s,Q}$ and $p_{r,n}^{s,Q}$ distinctly.

In the dual-frequency undifferenced and uncombined PPP model, it is assumed that $j$ satellites are simultaneously tracked by the receiver r. Equations (1) and (2) can be rewritten as:

$$\begin{bmatrix} p_{r,1}^{1,Q} \\ l_{r,1}^{1,Q} \\ \vdots \\ p_{r,2}^{j,Q} \\ l_{r,2}^{j,Q} \end{bmatrix} = \begin{bmatrix} -\mathbf{u}_r^{s,Q} & 1 & M_{\mathrm{w}} & \mathbf{K} & R_1 & R_2 \end{bmatrix} \begin{bmatrix} \mathbf{x} \\ d\bar{t}_r^Q \\ Z_{\mathrm{w}} \\ \bar{\mathbf{I}}_1^Q \\ \overline{\mathbf{N}}_1^Q \\ \overline{\mathbf{N}}_2^Q \end{bmatrix} + \begin{bmatrix} m_{r,1}^Q + \varepsilon_{r,1}^Q \\ M_{r,1}^Q + \xi_{r,1}^Q \end{bmatrix}, \tag{6}$$

with

$$\begin{cases} d\bar{t}_r^Q = dt_r + d_{r,\mathrm{IF}_{12}}^Q \\ \bar{\mathbf{I}}_1^{s,Q} = I_{r,1}^{s,Q} + \beta_{12}^Q \cdot \left(DCB_{r,P_1P_2}^Q - DCB_{P_1P_2}^{s,Q}\right) \\ \overline{\mathbf{N}}_1^{s,Q} = \lambda_1^Q \cdot \left(N_{r,1}^{s,Q} + b_{r,1}^{s,Q} - b_1^{s,Q}\right) + b_{\mathrm{IF}_{12}}^{s,Q} - b_{r,\mathrm{IF}_{12}}^{s,Q} + \beta_{12}^Q \cdot \left(DCB_{r,P_1P_2}^Q - DCB_{P_1P_2}^{s,Q}\right) \\ \overline{\mathbf{N}}_2^{s,Q} = \lambda_2^Q \cdot \left(N_{r,2}^{s,Q} + b_{r,2}^{s,Q} - b_2^{s,Q}\right) + b_{\mathrm{IF}_{12}}^{s,Q} - b_{r,\mathrm{IF}_{12}}^{s,Q} + \gamma_2^Q \cdot \beta_{12}^Q \cdot \left(DCB_{r,P_1P_2}^Q - DCB_{P_1P_2}^{s,Q}\right) \end{cases}, \tag{7}$$

where 1 expresses a unit column vector of receiver clock offsets $d\bar{t}_r^Q$, which has $2 \times j$ rows; $\mathbf{K}$ represents the unit column vector of ionospheric parameter $\bar{\mathbf{I}}_1^Q$, which the ingredient assigned to $p_{r,1}^{1,Q}$ is 1 and the factor assigned to $l_{r,1}^{1,Q}$ is $-1$; $R_1$ indicates the matrix of ambiguity parameters $\overline{\mathbf{N}}_1^{s,Q}$, the factor assigned to $p_{r,1}^{s,Q}$ is 0 and the factor assigned to $l_{r,1}^{s,Q}$ is 1; $R_2$ indicates the matrix of the ambiguity parameters $\overline{\mathbf{N}}_2^{s,Q}$, the factor assigned to $p_{r,2}^{s,Q}$ is 0 and the factor assigned to $l_{r,2}^{s,Q}$ is 1.

In this study, we only discuss the code multipath, and the comprehensive data handling strategies are stated in Table 1. The parameters $\overline{\mathbf{X}} = \left[ \mathrm{d}\bar{t}_r^Q, \overline{Z}_\mathrm{w}, \overline{\mathbf{I}}_1^{s,Q}, \overline{\mathbf{N}}_1^{s,Q}, \overline{\mathbf{N}}_2^{s,Q} \right]$ that are estimated with a Kalman filter are substituted into (1) to extract the code multipath as follows:

$$E\left[ m_{r,i}^{s,Q} \right] = P_{r,i}^{s,Q} - \mathbf{u}_r^{s,Q} \cdot \mathbf{R}_r^s - \mathrm{d}\bar{t}_r^Q - M_\mathrm{w} \cdot \overline{Z}_\mathrm{w} - \gamma_i^Q \cdot I_1^{s,Q}, \tag{8}$$

where $E[*]$ is defined as extraction operator and $\mathbf{R}_r^s$ is acquired from previous receiver position. It can be seen that after eliminating other modeled errors, the remaining code residuals only contain multipath errors and random noise.

**Table 1.** Data processing strategies.

| Items | Strategies |
|---|---|
| Observations | BDS: B1/B3 |
| Sampling rate | 30 s |
| Elevation cutoff | 7° |
| Parameter estimator | Kalman filter |
| Satellite orbits and clocks | WHU MGEX precise orbit (5 min interval) and clock (30 s interval) products |
| Carrier phase windup | Corrected using the external model |
| Tidal load | Corrected using the IERS convention model |
| Relativity effects, Earth rotation | Corrected using the external model |
| Satellite and receiver antenna Phase center | Corrected with igs14.atx |
| Slant ionospheric delays | Estimated as random-walk noise parameters ($0.01\mathrm{m}^2/\mathrm{s}$) |
| Tropospheric delays | The mapping function utilized for line of sight direction is global mapping function, zenith hydrostatic delays are corrected using the Saastamoinen model, zenith wet delays are estimated as random-walk noises ($10^{-7}\ \mathrm{m}^2/\mathrm{s}$) |
| Receiver clocks | Estimated as white noises |
| Phase ambiguities | Estimated as float constants |
| Station coordinates | Estimated as day constants |
| Stochastic model | Elevation-dependent weighting (prior variance as 0.003 and 0.3 m for code and phase observations) |

### 2.2. Multipath Analysis Method Based on Different Indicators

Multipath means that the satellite signal arrives at the antenna through multiple paths after being blocked, reflected and refracted by obstacles in the environment surrounding the receiver. The amplitude and phase of the multipath signal depend on the receiver's position as well as the environment. Therefore, existing studies generally argue that the code multipath can be modeled as a function of the corresponding satellite azimuth and elevation, unless there are changes in the nearby environment. However, even if the environment near the antenna remains unchanged in application, multipath errors exhibit characteristics related to the variations of satellite position. It is indispensable to explore the correlation between nadir angle with code multipath.

Ruan conducted the detailed modeling of satellite-induced multipath and proposes that the satellite-induced multipath should be established as functions that are relative to the nadir angle instead of the elevation angle [40]. In this study, we take the nadir angle into consideration. The satellite-induced variation of the multipath error can be well characterized by the nadir angle. This model will be proved next.

As shown in Figure 1, a triangle is formed by satellite S, station A and geocenter O, the earth is approximated as a sphere, and according to the Law of Sines:

$$\frac{sin(e_A + 90°)}{R_e + H_s} = \frac{sin(e_z)}{R_e + H_a} \tag{9}$$

where indicates $e_A$ and $H_a$ are the elevation angle and altitude of station A, respectively; $e_z$ represents the nadir angle of station A; $H_a$ represents the altitude of station A, $R_e$ indicates the earth's radius and $H_s$ represents the altitude of satellite S.

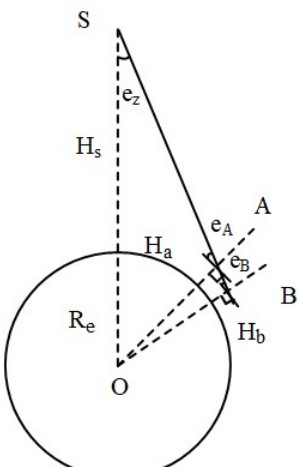

**Figure 1.** Geometrical relationship of satellite, earth and station.

For another station B on the LOS vector AS, the similar mathematic relationship can be formed as follows:

$$\frac{sin(e_B + 90°)}{R_e + H_s} = \frac{sin(e_z)}{R_e + H_b} \tag{10}$$

where indicates $e_B$ and $H_b$ are the elevation and altitude of station B, respectively.

Substituting (10) into (9), the equation can be expressed as:

$$\frac{sin(e_A + 90°)}{R_e + H_a} = \frac{sin(e_B + 90°)}{R_e + H_b} \tag{11}$$

It is clear that stations at different altitudes observe the same satellite at different elevation angles. We introduce the nadir angle as an independent variable to jointly model the satellite-induced multipath with receivers at different altitudes.

As mentioned before, the main parameters for each GNSS satellite include C/N0 to reflect the characteristics of multipath effects [14]. Different from SNR, C/N0 is irrelevant to the receiver's front-end bandwidth and represents the carrier power-to-noise power ratio that normalized to the unit bandwidth. The multipath signal is a composite signal formed by the reflection, refraction and diffraction components induced by obstacles in the surroundings. The multipath effect can modify the signal's amplitude and phase, causing the distortion of the original signal. Generally, the composite multipath signals can be expressed as:

$$S_m = A_d cos\varphi + A_i cos(\varphi + \Delta\varphi), \tag{12}$$

where $A_d$ and $A_i$ indicate the amplitudes of direct and indirect signals, individually; $\varphi$ represents the direct signal phase; and $\Delta\varphi$ represents the phase shift delayed by the indirect signal.

The C/N0 can reflect the composite signal's amplitude formed by superposing several multipath components in accordance with each phase. According to (12), the relationship is expressed as:

$$\text{C/N0}^2 = A_d^2 + A_i^2 + 2A_d A_i cos\Delta\varphi, \tag{13}$$

It is obvious that C/N0 represents the quality of received signal, and the multipath signal has a notable influence on C/N0. Therefore, it is reasonable to characterize the multipath signals with C/N0.

In this study, the azimuth angle, elevation angle, nadir angle and C/N0 are taken as multiple joint indicators to characterize the multipath.

## 3. Multipath Characterization with the AT-Conv-LSTM Network

### 3.1. AT-Conv-LSTM Network

In this study, we propose a multipath error modeling method based on AT-Conv-LSTM network. The algorithm flowchart is shown in Figure 2: (1) Raw BDS-3 observations collected from GNSS stations are stored in the database, and then undergo a post-processing procedure to proceed multipath extraction operations to be used as training data. (2) A sliding window size is set, and the AT-Conv-LSTM network is updated by incorporating the earlier model and incoming multipath errors, and the multipath errors are estimated in the current epoch. (3) The code multipath is corrected at the corresponding epoch based on the AT-Conv-LSTM network to mitigate the multipath.

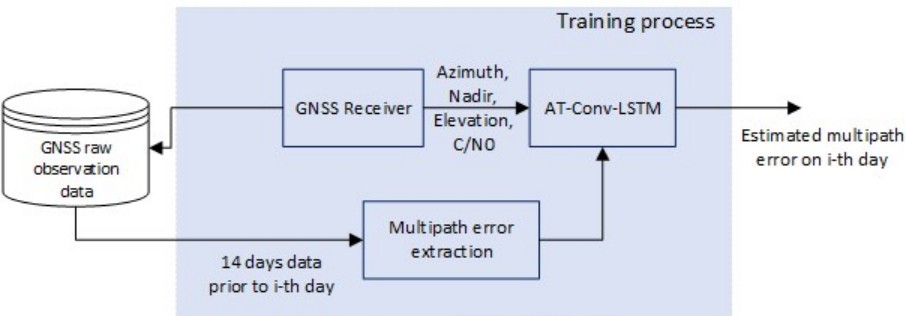

**Figure 2.** Multipath mitigation flowchart.

Since multipath errors are commonly considered to be random and nonlinear, it is imperative for a robust multipath model to possess the capability to accurately represent these intricate characteristics. Multipath errors are usually characterized via spatiotemporal correlation and periodicity. More specifically, the point of interest (POI) region of the multipath error is not only related to the multipath errors of its neighboring observation epoch but also its dependence on previous time. Moreover, the multipath errors also exhibit periodic repetitive patterns. In this paper, the AT-Conv-LSTM network is proposed to estimate the multipath errors. The proposed model comprises two Conv-LSTM modules integrated with attention mechanism, which can effectively utilize the spatiotemporal correlation and mitigate the multipath errors of BDS-3, as depicted in Figure 3.

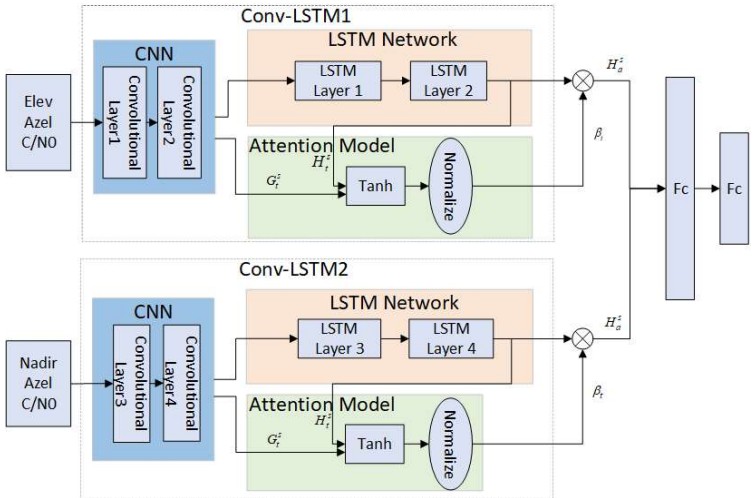

**Figure 3.** AT-Conv-LSTM network.

As mentioned earlier, the multipath is associated with the station environment, satellite position and receiver position. The multiple indicators that azimuth angle, elevation angle,

nadir angle and C/N0 are taken to characterize the multipath significance. The multipath error can be denoted at $i$th satellite vary with the multiple indicators as:

$$m_{r,i}^{s,Q} = f(az_i, elev_i, nadir_i, f_i, CNR_i),\tag{14}$$

where $az_i$, $elev_i$ and $f_i$ are the azimuth angle, elevation angle and frequency; $nadir_i$ represents the nadir angle; and $CNR_i$ represents the C/N0 of $i$th satellite.

Hence the inputs of the first Conv-LSTM module are elevation angles and azimuth angles represent the variation related to receiver's surroundings and position, and the inputs of the second Conv-LSTM module are nadir angles and azimuth angles represent the variation related to satellite's position.

The core component of the proposed network is the Conv-LSTM module, encompassing two convolutional layers and two LSTM layers. The C/N0, nadir angle and azimuth angle of the $m$th satellite on the epoch $n$ can be represented as $X_n^m = [az_n^m, nadir_n^m, CNR_n^m]$. Subsequently, we aggregate the historical angles from its neighboring locations (total n epochs) as follows:

$$X_t^s = \begin{bmatrix} X_1^s \\ X_2^s \\ \vdots \\ X_n^s \end{bmatrix} = \begin{bmatrix} X_1^1 & X_2^1 & \cdots & X_n^1 \\ X_1^2 & X_2^2 & \cdots & X_n^2 \\ \vdots & \vdots & \ddots & \vdots \\ X_1^m & X_2^m & \cdots & X_n^m \end{bmatrix},\tag{15}$$

where $s = 1, 2, \ldots, m$ symbolizes the count of satellites and $t = 1, 2, \ldots, n$ symbolizes the count of epochs.

The matrix $X_t^s$ at each epoch $t$ is subjected to a one-dimensional convolution process in order to capture the spatial feature. A sliding filter is used to capture the local perceptual domain using a one-dimensional convolution kernel filter. The following illustration shows how the convolution kernel filter works:

$$Y_t^s = \sigma(w_s * X_t^s + b_s),\tag{16}$$

where $w_s$ represents the filter weights, $b_s$ represents the bias, symbol * defined as the convolution operation, $\sigma$ represents the activation function and $Y_t^s$ expresses the convolutional layer output. The network uses the tanh activation function. The aforementioned process promotes the extraction of the spatial feature from the neighboring observation regions.

To enhance optimize the efficiency of the deep neural network, the most common approach involves augmenting the model through the addition of layers. By incorporating multiple LSTM layers into the network, this study enhances the network's capacity to adapt the multipath errors. After being extracted through two convolutional layers, the spatial features are inputted into the stacked LSTM network. Through the stacking of LSTM layers, each subsequent layer in the stack receives the hidden state served as the input of preceding layer. The LSTM's mathematical equation with different cell states is given below:

$$\begin{aligned} i_t &= \sigma\left( w_i \left[ h_{t-1}^s, X_t^s \right] + b_i \right) \\ f_t &= \sigma\left( w_f \left[ h_{t-1}^s, X_t^s \right] + b_f \right) \\ \widetilde{C} &= \tanh\left( w_c \left[ h_{t-1}^s, X_t^s \right] + b_c \right) \\ C_t &= f_t \circ C_{t-1} + i_t \circ \widetilde{C} \\ o_t &= \sigma\left( w_o \left[ h_{t-1}^s, X_t^s \right] + b_o \right) \\ H_t^s &= o_t \circ \tanh(C_t) \end{aligned} \tag{17}$$

where $i_t$ represents the input of the LSTM layer on epochs $t$, $i_t$, $f_t$, $o_t$, which indicate the input gate, the forget gate and output gate at epoch $t$, respectively; $\circ$ expresses the Hadamard product; and $w$ and b represent the weights and biases of the network, respectively. Finally, we obtain the spatiotemporal feature $H_t^s$ for time step $t$.

*3.2. Attention Mechanism Considering Multiple Indicators*

The introduction of the attention mechanism aims to investigate the intrinsic features of the sequence and enhance the effectiveness of information handling. It enables models to assign different weights to different positions within the input sequence, enabling them to concentrate on the most significant components while processing each sequence element.

After two convolution layers, the spatial feature has been extracted from the input matrix $X_t^s$, including the C/N0, nadir and azimuth angles. $G_t^s$ is denoted the convolutional layer 2 output. As is widely recognized, the multipath also demonstrates temporal correlations in adjacent epochs. LSTM is usually employed to uncover hidden temporal features in a time series. Therefore, after the processing of spatial information through the two convolutional layers, the output is subsequently linked to the LSTM network. Thus, we obtain the spatiotemporal feature $H_t^s$.

The observation data are not continuous in the time domain due to the limited visibility time of MEO and GEO satellites for a specific observation station. In order to address this issue, we introduce an attention mechanism to calculate the important score of each Conv-LSTM output. Through the attention mechanism, we can obtain the estimated multipath that automatically assigns a different importance score to each visible epoch.

The Conv-LSTM output at epoch $t$ is calculated by combining the outputs of the CNN and LSTM module with weighting coefficients, as demonstrated below:

$$s_t = v_s^T tanh(w_h G_t^s + w_l H_t^s)$$
$$\beta_k = \frac{exp(s_k)}{\sum_{k=1}^{n+1} exp(s_k)} ,$$
$$H_t^a = \sum_{k=1}^{n+1} \beta_k H_{t-(k-1)}^s, \tag{18}$$

where $s_t$, $\beta_k$, and $H_t^a$ are, respectively, the importance score of each input part, the attention value and the output at epoch $t$. $w$ and v are the weights. Ultimately, the spatiotemporal feature $H_t^a$ for time step $t$ is obtained.

The same structure as the other Conv-LSTM will not be explained here. Subsequently, all these features are combined into a feature vector, which is then fed into two regression layers for estimating purposes. The objective function of regression involves a loss function that calculates the mean squared error of the estimated multipath errors.

*3.3. Model Training and Evaluation*

Within the proposed model, the optimization of model parameters is achieved by employing the Adam optimization algorithm, which enables adaptive adjustment with the learning rate.

In order to provide a quantitative evaluation of the estimated accuracy of, the root mean square error (RMSE) and mean absolute error (MAE) are selected to calculate the difference between the real multipath error extracted in Section 2.1 and the estimated value. The following is the mathematical equation of RMSE and MAE:

$$RMSE = \sqrt{\frac{1}{n}\sum_{t=1}^{n}(m_t - \hat{m}_t)^2}, \tag{19}$$

$$MAE = \frac{1}{n}\sum_{t=1}^{n}|m_t - \hat{m}_t|, \tag{20}$$

where $n$ represents sampling points number, $m_t$ represents the extracted multipath in previous section and $\hat{m}_t$ is estimated value of multipath for $i$th sampling point. Moreover, as the RMSE and MAE values approach zero, the network's estimated results become closer to the actual multipath error, indicating better performance.

## 4. Results

### 4.1. Data Description

The BDS global PNT services consists of the experiment system (BDS-1), the regional system (BDS-2) and the global system (BDS-3) and reached full operational status on July 31, 2020 [41]. The BDS offers high-precision, real-time services to users worldwide. They are made up of a constellation of Medium Earth Orbit (MEO) (C19–C30, C32–C37, C41–46), Inclined Geosynchronous Orbit (IGSO) (C31, C38–C40) and Geosynchronous Earth Orbit (GEO) (C59, C60) satellites, which is different from GPS, GLONASS and Galileo. In addition to the antiquated B1l and B3l broadcasts that were carried over from the BDS-2 satellites, the BDS-3 satellites broadcast a variety of new open service signals. The frequencies, wavelengths and chip rates of BDS-3 signals are stated in Table 2. In our study, we analyzed the BDS dual-frequency observations (B1/B3) from the International GNSS Service (IGS) MGEX station for 16 days from DOY 160-176, 2021. The multi-frequency GNSS station, namely JFNG, is capable of tracking BDS, as shown in Table 3. The multi-frequency GNSS station, namely JFNG, is capable of tracking BDS as shown in Figure 4. And the detail station information is stated in Table 3.

**Table 2.** BDS-3 signal characteristics.

| BDS-3 System | B1I | B1C | B2a | B2b | B3I |
|---|---|---|---|---|---|
| Frequency (MHz) | 1561.098 | 1575.420 | 1176.450 | 1207.140 | 1268.520 |
| Chip Rate (Mcps) | 2.046 | 1.023 | 10.23 | 10.23 | 10.23 |
| Wavelength (cm) | 19.20 | 19.03 | 25.48 | 24.83 | 23.63 |

**Table 3.** Station information.

| Items | Strategies |
|---|---|
| Station Name | JFNG |
| Localization | China |
| Latitude | 30.51557° |
| Longitude | 114.49102° |
| Receiver | TRIMBLE ALLOY—6.20 |
| Antenna Type | TRM59800.00 |
| Constellations | GPS + GLO + GAL + BDS + QZSS + IRNSS + SBAS |

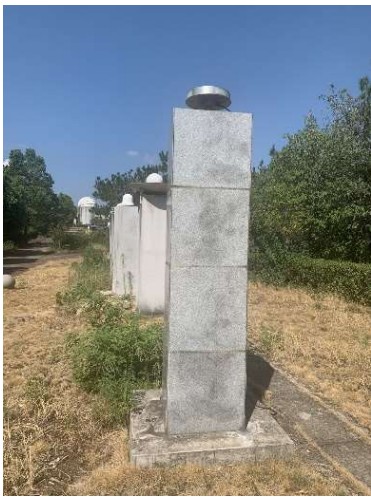

**Figure 4.** JFNG station.

### 4.2. Code Multipath Analysis

The BDS-3 MEO satellite (C22) and IGSO satellite (C38) are selected as examples for the analysis next, the multipath is extracted using the multipath analysis method

in equation (8). Figure 5 shows the BDS-3 MEO satellite (C22) and IGSO satellite (C38) code multipath on B1I and B3I frequency. When the satellite just enters the visible range, the elevation is small, while the lower elevation angle leads to a lager code multipath and bias of dual frequency. This phenomenon may be due to the fact that signals from satellites with low elevation angles have a higher probability of occurrence. Compared with MEO satellites, the multipath of IGSO satellites with dual frequency show more obvious differences, and the maximum differences can reach the meter level. The bias of each code multipath and the difference between the multipath on dual frequency can seriously affect the accuracy of positioning. The low and high-frequency parts can be found in the multipath, which definitely reduces the positioning accuracy. Figure 6 shows number of visible BDS-3 satellites on DOY 176 and the corresponding dilution of precise (DOP). The number of tracked satellites varied between 6 and 11. The vertical-DOP (VDOP) and the horizontal-DOP (HDOP) values vary from 0.5 to 3, showing that BDS-3 is capable of delivering accurate positioning services independently.

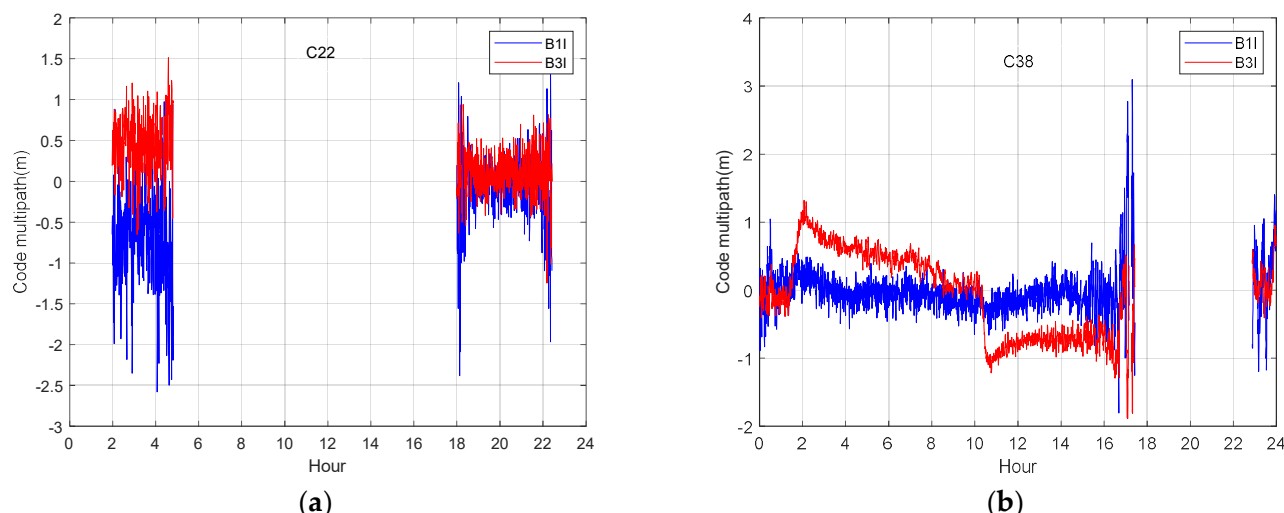

**Figure 5.** Code multipath at B1I and B3I of station JFNG on DOY 176,2021. (**a**) Code multipath at two frequencies of MEO satellite; (**b**) code multipath at two frequencies of IGSO satellite.

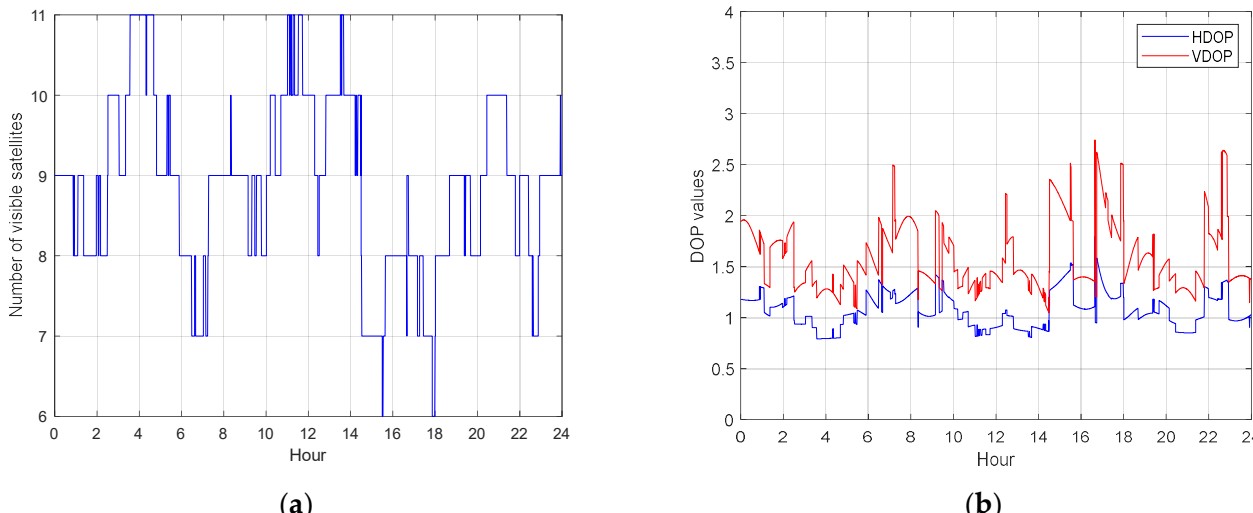

**Figure 6.** BDS-3 visible satellites on DOY 176. (a) number of BDS-3 visible satellites (b) HDOP and VDOP of BDS-3 satellites.

Figure 7 shows the probability density of the code multipath. Most code multipath range from −2 to 2 m, with almost 95% ranging from −1 to 1 m. The code multipath is

obviously larger in the low elevation angle region. When the altitude angle is lower than 30°, the code multipath is generally larger than 1 m and the code multipath of signals in the high elevation regions are generally smaller than 1 m; therefore, the code multipath is highly correlated with the elevation angles. The most elevation-dependent effects are eliminated above 35° elevation [42]. This phenomenon occurs because most of the external effects can be eliminated in the signal accuracy analysis when the satellite altitude is higher than 35°. It is noted that there is a constant bias in multipath, actually this is caused by the unmodeled error (e.g., hardware delay) [14]. Fortunately, the constant bias present in the multipath will not influence multipath analysis, and it is reasonable to ignore the bias.

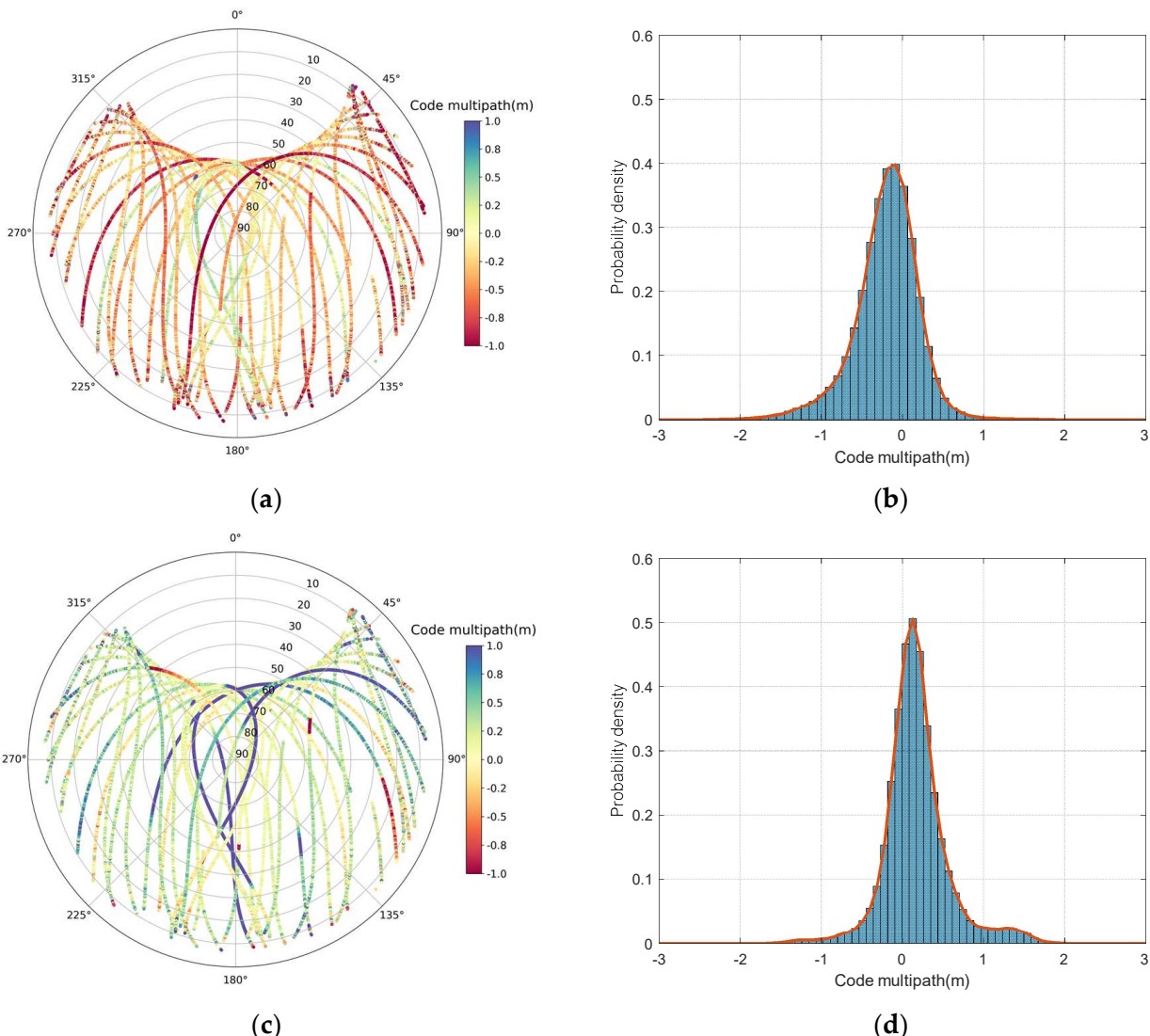

**Figure 7.** BDS-3 multipath sky map and histogram of code multipath. (**a**) B1I multipath sky map; (**b**) histogram of B1I code multipath; (**c**) B3I multipath sky map; (**d**) histogram of B3I code multipath. The red line in right panels is the envelope of the code multipath distribution.

Figure 8 shows the BDS-3 MEO satellite (C22) and IGSO satellite (C38) C/N0 on the B1I and B3I frequencies with respect to elevation angle. At lower elevations, the C/N0 is small, which corresponds to a large code multipath in Figure 5. The C/N0 value decreases significantly and the code multipath becomes large and divergent, particularly when the elevation angle is below 10°. This justifies the rationality of characterizing the variation of code multipath with C/N0.

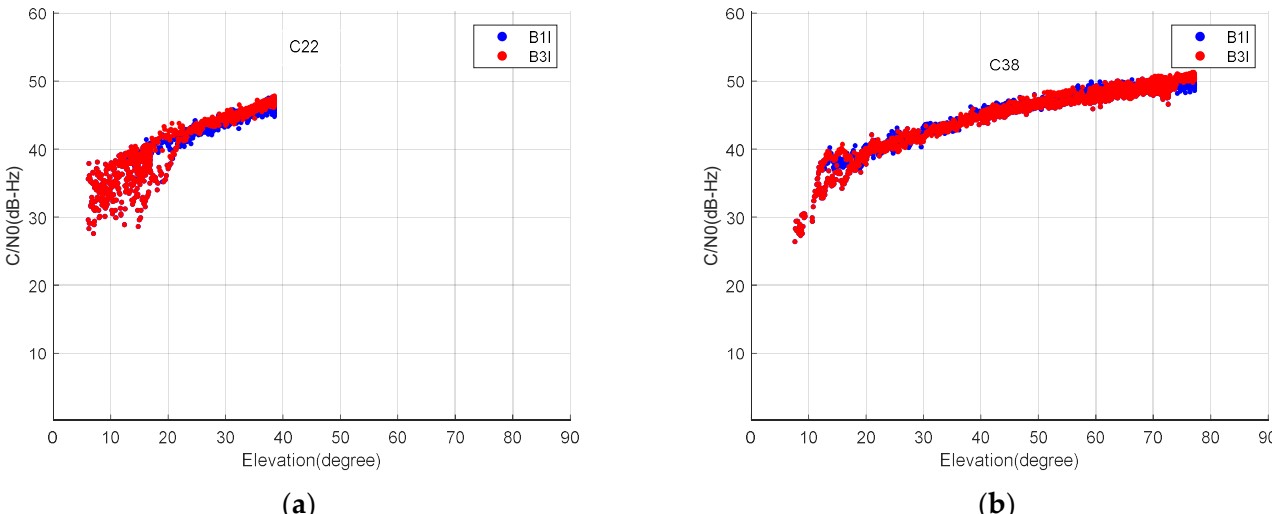

**Figure 8.** C/N0 of B1I and B3I of MEO satellites (**a**) and IGSO (**b**) with respect to elevation angle.

### 4.3. Correlation Analysis of Nadir Angles and Code Multipath

Figure 9 demonstrates that the variation of the elevation angle of MEO and GEO/IGSO satellites to the receivers at different elevations with the nadir angle. Along with the growth in elevation angle, the range of the observable nadir angle increases. The elevation angle changes with the station altitude for the same nadir angle, and as the nadir angle increases, the difference in elevation angle at different latitudes becomes more noticeable. For example, if the nadir is 13.21° (H = 0 km, elevation = 0°), the elevation angles are 3.21°, 7.15°, 10.08° and 21.99° when the station is located at 10 km, 50 km, 100 km and 500 km, respectively. In conclusion, extra errors will be introduced into multipath error correction using elevation angle as the feature, according to previous studies.

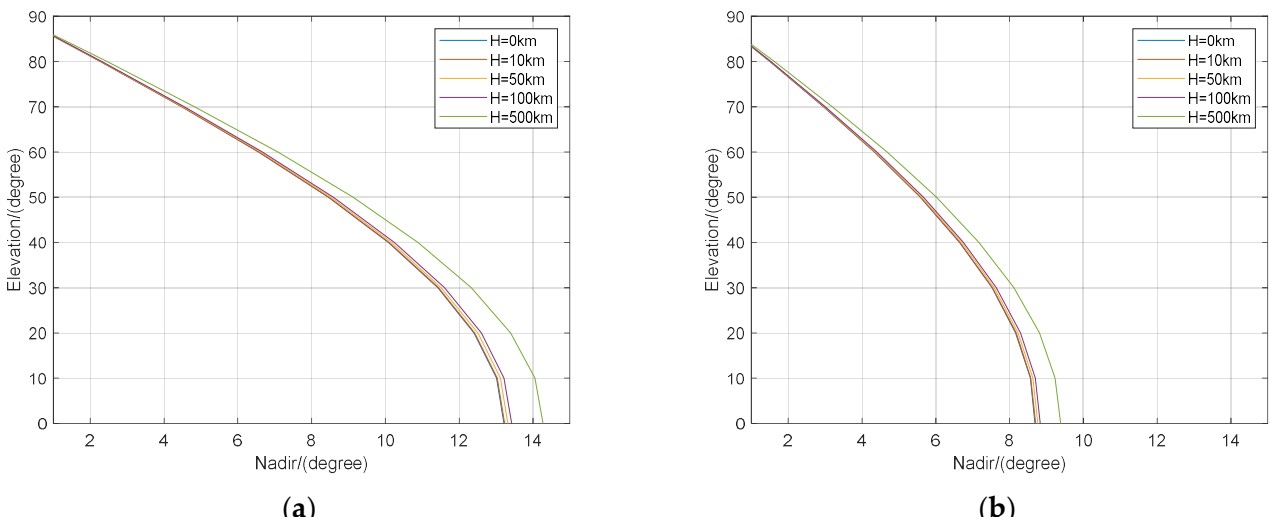

**Figure 9.** Elevations of MEO satellites (**a**) and GEO/IGSO (**b**) with respect to receivers at different altitude as a function of nadir angle.

From Figure 10, when the elevation angle exceeds 30°, the code multipaths of C22 are between −0.5 and 0.5 m and those of C38 are between −1 and 1 m, and when the elevation angle is smaller than 30°, the code multipaths are obviously increased. It can be concluded that the smaller the elevation and nadir angles are, the larger the code multipath and the more discrete distribution are. These results demonstrate the correlation of code multipath with the elevation and nadir angle. Moreover, the relationship between the elevation angle and nadir angle exhibits a nonlinearity that can clearly be seen from the code multipath of

C38, which verifies that the nadir angle rather than the elevation angle should be used as the independent variable for multipath error modeling.

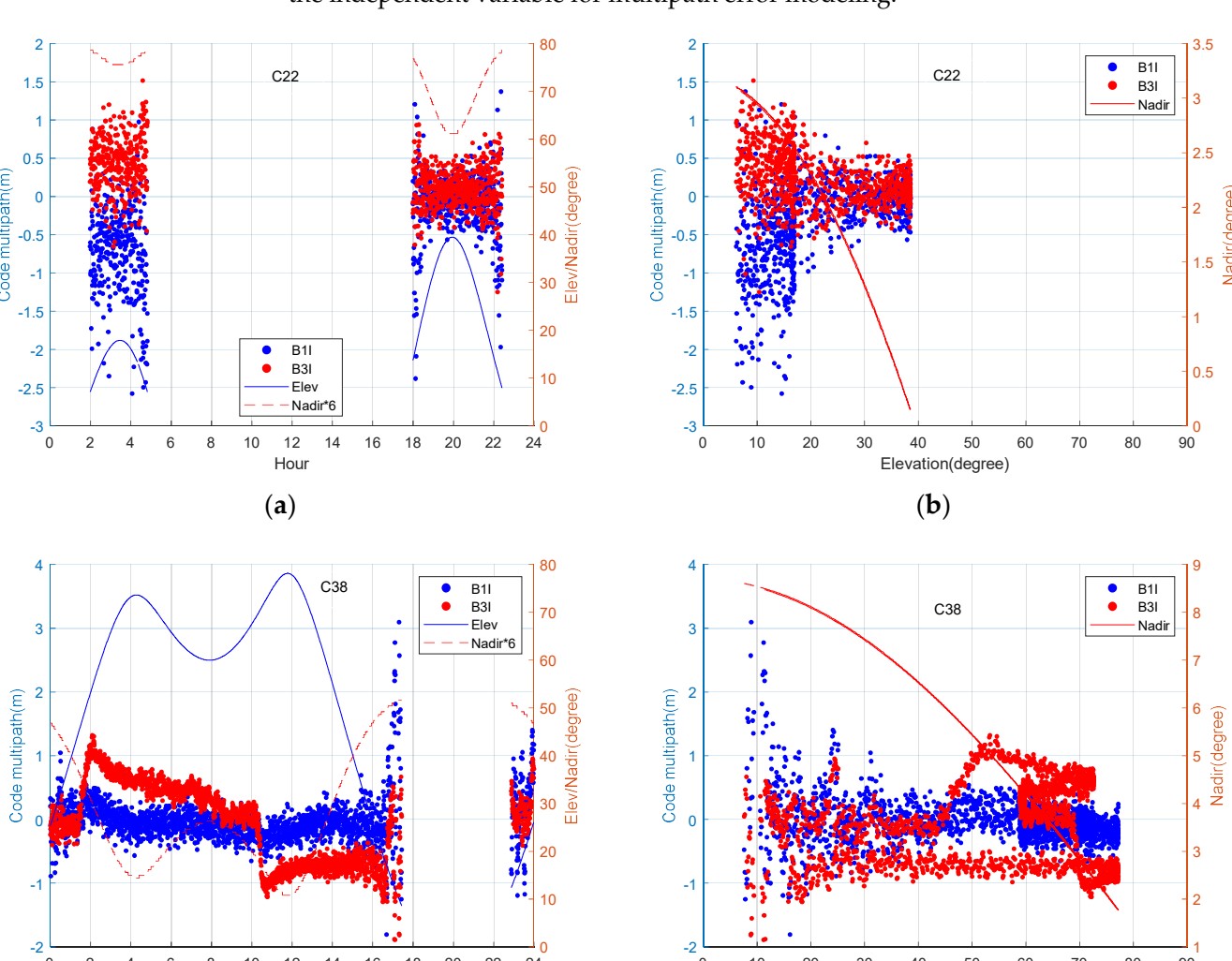

**Figure 10.** (**a**–**d**) Variation of BDS satellites multipath and nadir angles with respect to epoch or elevation angle. The amplified values of nadir angles are used to make the relationship clear.

### 4.4. Comparison of AT-Conv-LSTM with Other Methods

To assess the effectiveness of the multipath mitigation method based on the AT-Conv-LSTM network, we used the data from DOY 160–175 of JFNG station 2021 for training and those from DOY 176 for testing. The left panel in Figure 11 illustrates the uncorrected code multipaths of MEO (C22) and IGSO (C38) before the multipath correction and the multipath errors predicted by AT-Conv-LSTM, respectively. The right panel in Figure 11 illustrates the bias of the uncorrected code multipath minus the predicted multipath error. The MAE of the uncorrected multipath is 0.3322, and the MAE of multipath corrected by the AT-Conv-LSTM decreases to 0.0681. As can be seen, our method can effectively mitigate the code multipath, and can essentially correct the fluctuating low-frequency components. The corrected multipath errors are analogous to the white noise series, signifying that the AT-Conv-LSTM network is able to mitigate the multipath effectively.

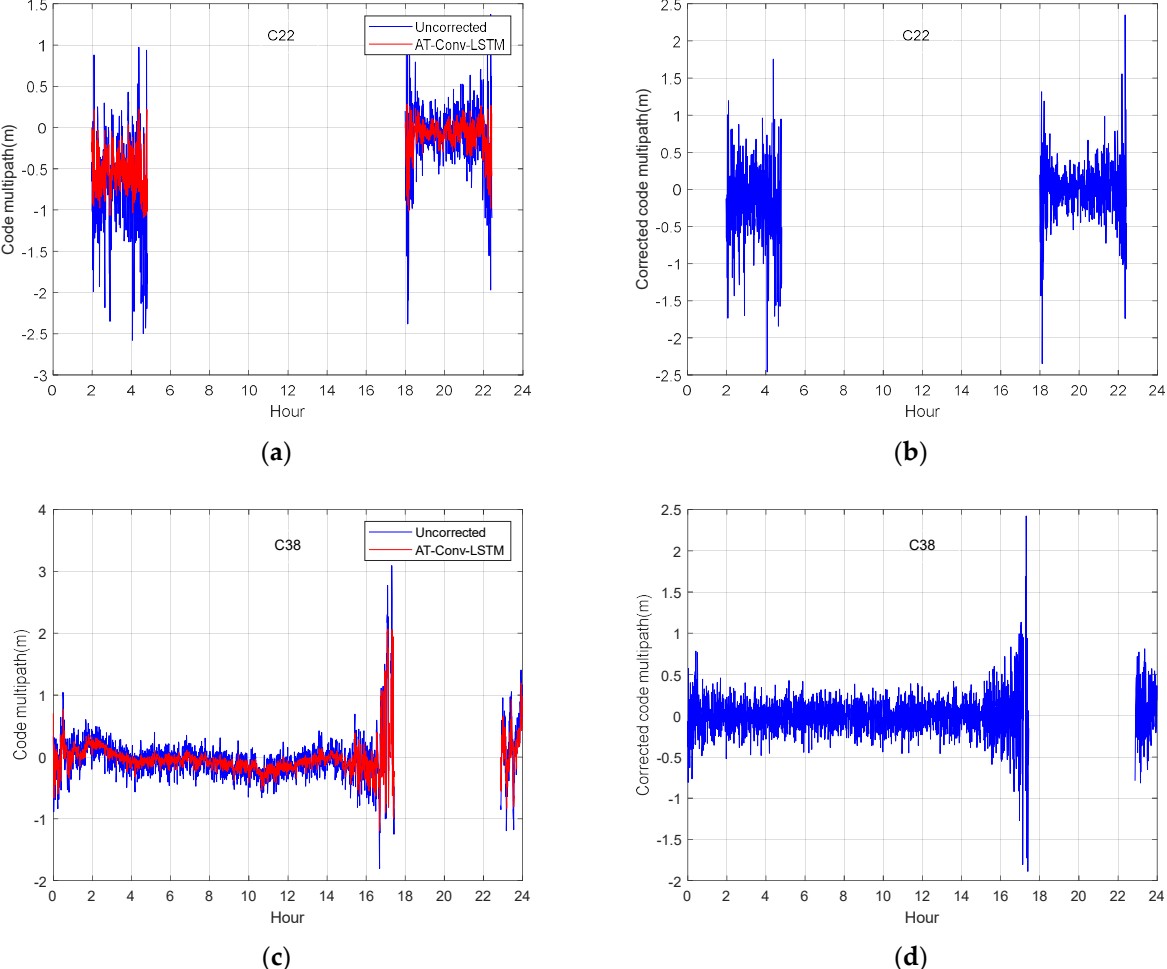

**Figure 11.** MEO satellite C22 multipath before (**a**) and after correction (**b**); IGSO satellite C38 multipath before (**c**) and after correction (**d**).

To confirm the effectiveness of multipath mitigation using the AT-Conv-LSTM network, we conducted comparative experiments with SF, MHM and T-MHM for comparison. The surrounding environments of the stations remain unchanged. The ORTs of BDS satellites are calculated on an individual basis in advance of the SF method. The multipath models were first established through the code multipath of the corresponding days using low-pass-filters, then were removed from the code multipath of next day. Figure 12 illustrates that the multipath errors on DOY 166 and DOY 173 of C12 satellite has a strong temporal correlation, the correlation coefficient is 0.82 and the multipath error of C38 satellite represents a strong temporal correlation in three days DOY 171, DOY 172 and DOY 173, the correlation coefficient is 0.79. Considering the differences between the IGSO and MEO orbits, we correct the multipath errors of the IGSO and MEO satellites individually. In this study, our analyses primarily focused on the ORTs of IGSO satellites (such as C38) and MEO satellites (such as C22). We extract the multipath errors of the IGSO and MEO satellites by using the data of the previous day and the previous seven days, respectively. The time advances of different orbital satellites are computed through code multipath correlation, in order to conduct multipath mitigation experiments in observation domain. As claimed by the analyses above, the ORT of GEO and IGSO satellites is around 86,165 s, while the ORT of MEO satellites is around 84,697 s and six days.

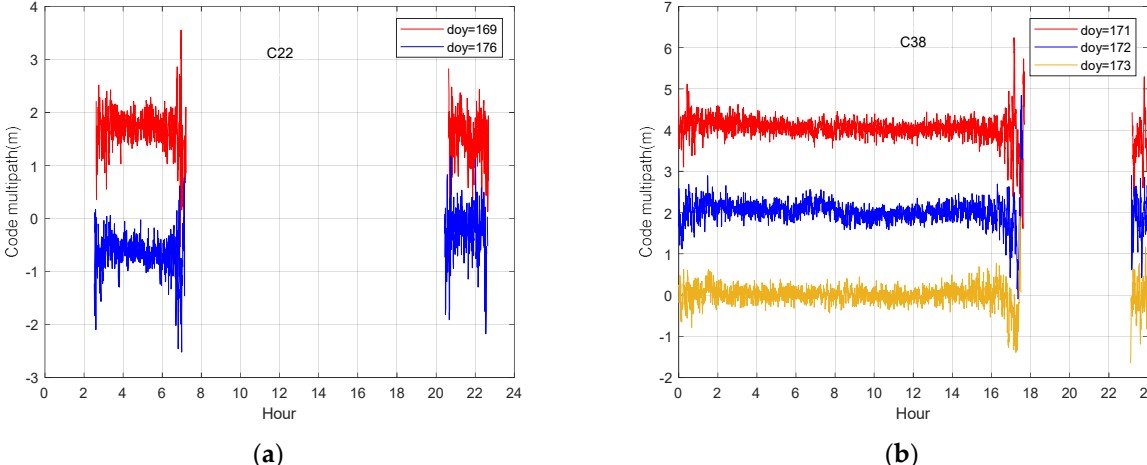

(**a**)　　　　　　　　　　　　　　　　　(**b**)

**Figure 12.** Comparison of the multipath errors for C22 (**a**) and C38 (**b**). The vertical 2 m offset was applied to the code multipath sequences for illustration purposes.

The MHM method for multipath error mitigation involves dividing the multipath into sky grids of certain sizes determined by the azimuth and elevation angles. Next, the average multipath from all satellites in the certain grid is used to construct a multipath error correction value table. We used the DOY 160–175 data to construct the MHM grids, as shown in Figure 13. The multipath curve obtained from MHM exhibits similarities to rectangular waves and is characterized by a deficiency in high-frequency information. Afterwards, T-MHM as an improved method is proposed to describe the multipath spatial distribution per grid specifically. The multipath code is divided into a grid in the sky, with dimensions determined by the azimuth and elevation angles. This grid is used to conduct trend-surface analysis on the multipath within each specific grid. The resulting trend-surface fitting coefficients are then stored. We filtered and denoised the multipath data from different satellites in the next experiments and removed the outliers based on the 3-sigma principle, then divided the grid points into sizes $1° \times 1°$ consistent with the MHM. Finally, a linear function is adopted to fit the trend surface, and the coefficients are stored to construct the correction table.

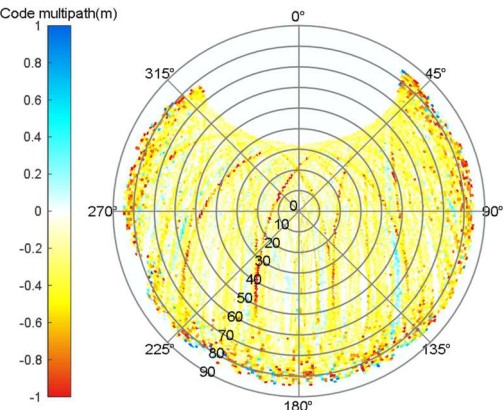

**Figure 13.** Sky map of the MHM grid.

The DOY 176 satellites data with two types of orbits such as MEO C22 (upper) and IGSO C38 (bottom) are used for comparison in Figure 14. Compared the estimated multipath from our proposed method with the uncorrected code multipath and SF, MHM and T-MHM model in Figure 14. The upper panel illustrates that other methods fit well for low-frequency fluctuations, but poorly for more obvious high-frequency fluctuations, especially in C22 satellites. However, our proposed method is most consistent with the variation trend of the uncorrected code multipath series.

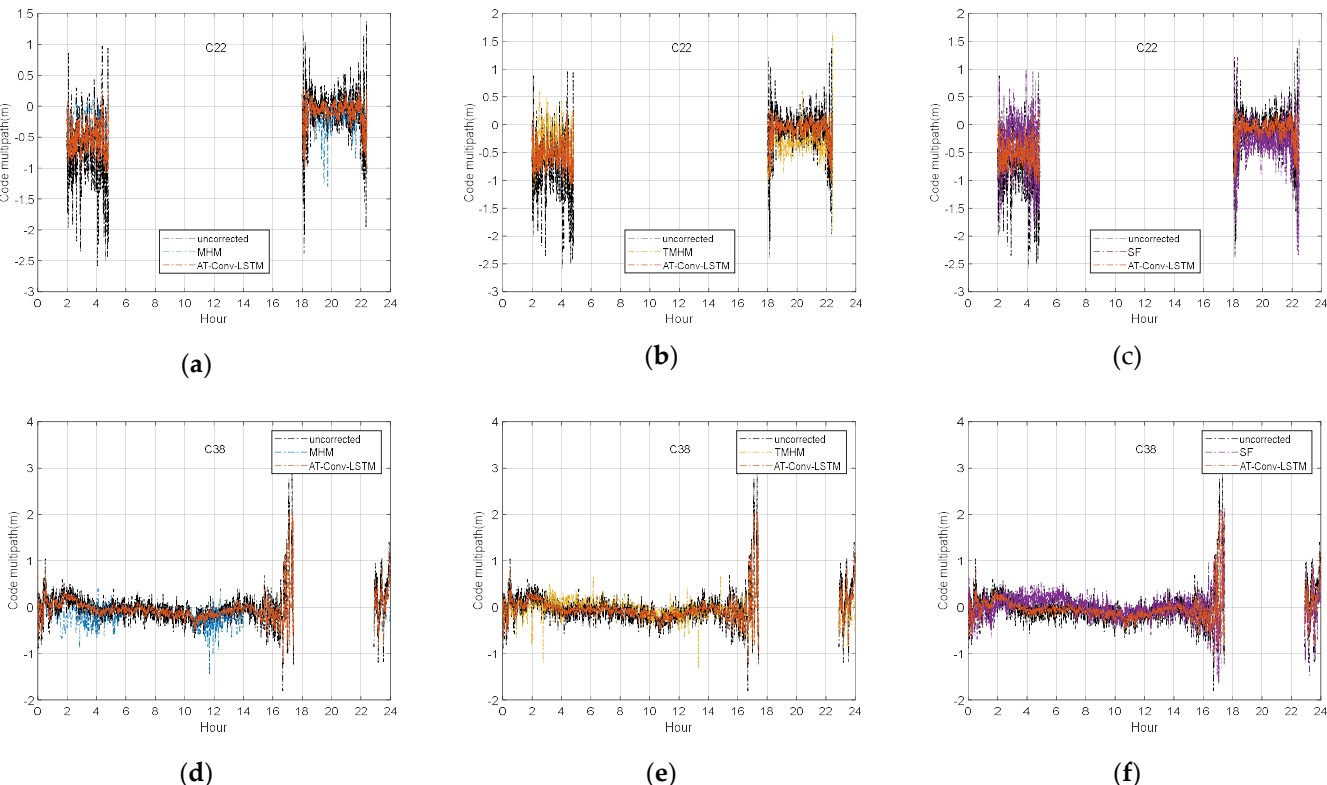

**Figure 14.** The time series of the C22 MEO satellite uncorrected code multipath and the AT-Conv-LSTM-estimated code multipath compared with MHM (**a**), T-MHM (**b**) and SF (**c**). The time series of the C38 IGSO satellite uncorrected code multipath and AT-Conv-LSTM-estimated code multipath compared with MHM (**d**), T-MHM (**e**) and SF (**f**).

The experiment used 16 days of data for MHM modeling. The substandard quality of observation data on certain days and the limited fitting ability of the linear function within the grid points can affect the MHM and TMHM modeling. And the inaccurate estimation of the ORTs of different satellites may affect the performance of the SF. All these reasons could affect the effect of the other three methods for the multipath correction. As shown in the comparison results in Table 4, the other methods have some improvements in the MAE value, but the improvement in the RMSE is not obvious. Our method reduces the MAE value of code multipath from 0.3322 to 0.0681 for MEO satellite C23 and from 0.0614 to 0.0241 for IGSO satellite C39, and the MAE correction of code multipath is improved by about 70%. The code multipath can be corrected to a centimeter-level bias near 0. The corrected code multipath exhibits a reduction in magnitude by one order compared to the original uncorrected code multipath. And there is a corresponding improvement in RMSE, which is reduced from 0.5596 to 0.4747 in MEO satellite C23 and from 0.2766 to 0.2386 in IGSO satellite C39, and the RMSE of the code multipath is improved by about 13%. Due to space limitations, we will only use the B1I frequency point as an example in the above performance analysis and calculate the correction enhancement of the code multipath MAE using AT-Conv-LSTM, respectively. Taking C22, C23, C36, C38 and C39 as examples, the correction of the code multipath MAE can reach more than 60%.

**Table 4.** MAE and RMSE values of code multipath with and without multipath model correction in Figure 14.

| PRN | Uncorrected | | MHM-Corrected | | T-MHM-Corrected | | SF-Corrected | | AT-Conv-LSTM-Corrected | |
|---|---|---|---|---|---|---|---|---|---|---|
| | MAE (m) | RMSE (m) | MAE (m) | RMSE (m) | MAE (m) | RMSE (m) | MAE (m) | RMSE (m) | MAE (m) | RMSE (m) |
| C22 | 0.3322 | 0.5596 | 0.1581 | 0.6154 | 0.0143 | 0.5731 | 0.0050 | 0.6608 | 0.0681 | 0.4447 |
| C23 | 0.3322 | 0.5596 | 0.1797 | 0.6175 | 0.0143 | 0.5731 | 0.0050 | 0.6608 | 0.0681 | 0.4747 |
| C36 | 0.3620 | 0.3809 | 0.1827 | 0.4225 | 0.0529 | 0.2952 | 0.0587 | 0.5641 | 0.0812 | 0.3615 |
| C38 | 0.0340 | 0.3364 | 0.1106 | 0.3932 | 0.0020 | 0.3588 | 0.0340 | 0.3364 | 0.0480 | 0.4118 |
| C39 | 0.0614 | 0.2766 | 0.0505 | 0.3501 | 0.0191 | 0.2686 | 0.0447 | 0.3514 | 0.0241 | 0.2386 |

## 5. Discussion

The current multipath analysis methods are limited to a single indicator representation. However, this cannot fully describe the characteristics of the multipath. First, we analyzed the strong correlation between code multipath and nadir angles, elevation angles, azimuth angles and C/N0 in "Section 4.2. Code Multipath Analysis". In "Section 4.3 Correlation Analysis of Nadir Angles and Code Multipaths", we explained the nonlinear relationship between nadir angles and elevation angles. Our analysis suggests that nadir angles, elevation angles, azimuth angles and C/N0 should be jointly considered as characteristic indicators of code multipaths. There is a constant bias in multipaths, caused by the unmodeled error (e.g., hardware delay), which can be ignored due to its stability. And the multipath is caused by lots of factors, such as the reflection, diffraction and obstruction of signals by local obstacles. In the future, we might work on a more accurate multipath analysis method to adapt different types of multipath.

The AT-Conv-LSTM network spatial features of elevation angles, nadir angles, azimuth angles and C/N0 through convolutional layers and extracts temporal features from the output sequence of the convolutional layers through LSTM. An attention mechanism is introduced to automatically allocate greater weights to the visible periods of satellites. These modules enable AT-Conv-LSTM to better extract spatiotemporal features. In our simulation, the sliding window size cannot be set too small. Since the variation of elevation, azimuth and zenith angles between adjacent epoch elements is inconspicuous, a small sliding window size will result in the inability to adequately extract the temporal features. The sliding window size is configured as 120 points for a 1 h observation data with 30 s sampling intervals in our study. In addition, a dropout layer is added to prevent the network from overfitting, and the dropout rate is set to 0.6. From Figure 14 and Table 4, it can also be seen that the multipath correction results of AT-Conv-LSTM network can correct the code residuals to a deviation of around zero, which is one degree lower than original uncorrected code residuals and also shows improvement compared to the other three methods. Furthermore, we aim to characterize the phase multipath error with the elevation angle, azimuth angle, nadir angle and C/N0 value as multiple indicators. Specifically, a more suitable deep learning model can be further investigated to mitigate the phase multipath error effectively.

## 6. Conclusions

The multipath, which is widely recognized as the most challenging remaining error, imposes constraints on the accuracy of GNSS positioning. Despite efforts to develop precise error models, the multipath remains a significant source of error that hinders high-precision positioning. This study focuses on extracting the multipath error from the raw BDS-3 code observations using the undifferenced and uncombined PPP model. Firstly, we extracted the multipath error from the raw BDS-3 code observations based on the undifferenced and uncombined PPP model. Since the amplitude and phase of multipath signal rely on the position of satellite and receiver in addition to the environment, the correlation among

the multipath error and elevation, nadir and azimuth angle is analyzed. Correspondingly, we analyzed the non-linear relationship between elevation and nadir angles. Therefore, azimuth angle, elevation angle, nadir angle and C/N0 are taken as multiple indicators to characterize the multipath significance. Furthermore, an AT-Conv-LSTM network is proposed to exploit the temporal correction from the multiple indicators' changing pattern over time and exploit the spatial correction from the multiple indicators' changing pattern over angles. Thus, our method can maximize the temporal and spatial repeatability of the multipath for real-time multipath mitigation. And the proposed method takes into account the spatial distribution of multipath without requiring ORTs, which can also correct both low and high-frequency components of multipath errors. Finally, our method significantly decreases multipath MAE and RMSE in comparison to SF, MHM and TMHM. Moreover, it has the capability to correct code multipath with a deviation at the centimeter level. Therefore, the proposed AT-Conv-LSTM network could mitigate the multipath efficiently and will be of broad practical value in the fields of standard positioning service and high-precision positioning.

**Author Contributions:** Conceptualization, J.S. and J.W.; methodology, J.S. and Z.T.; software, J.S.; validation, J.S., J.W., Z.T. and C.Z.; formal analysis, Z.T.; investigation, J.S. and Z.T.; writing—original draft preparation, J.S. and Z.T.; writing—review and editing, J.S., C.Z. and J.W.; visualization, C.Z.; supervision, J.W. and Z.T. All authors have read and agreed to the published version of the manuscript.

**Funding:** This research was funded by the National Natural Science Foundation of China (Nos. 62171191 and 62271223).

**Data Availability Statement:** The GNSS raw observation data was available at https://cddis.nasa.gov/archive/gnss/ (accessed on 2 August 2023).

**Conflicts of Interest:** The authors declare no conflict of interest.

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
