# Peer review of "Characterization of BDS Multipath Effect Based on AT-Conv-LSTM Network"

_remotesensing, doi:10.3390/rs16010073_

Round 1
Reviewer 1 Report
Comments and Suggestions for Authors
In this paper, multiple indicators including nadir angles, elevation angles, azimuth angles, and C/N0 are introduced to characterize the code multipath. Additionally, the AT-Conv-LSTM network is proposed to estimate the code multipath with multiple indicators, and can fully exploit the spatiotemporal correlations of code multipath. The proposed method performs superior to the state of the arts and can widely apply to the standard positioning service and high-precision positioning. The article is basically logical and conforms to the subject of this journal. I think it can be received after minor revision.
1. The significance of this paper is not expound sufficiently. The authors use AT-Conv-LSTM network because it can fully exploit the spatiotemporal correlations of code multipath, this advantage should be depicted more explicit in the Conclusions sections.
2. More clarification is required for the AT-Conv-LSTM, such as the activation function used in this paper.
3. The background research on deep learning is not extensive enough. Is it possible to use another network framework? Representative studies should be supplemented in INTRODUCTION.
4. Line 144, equation (3) gives the ionosphere-free combination, and equations (1) and (2) also give the ionospheric delay. I think this is contradictory without a reasonable explanation. Please revise after careful consideration.
5. I suggest adding pictures of the surrounding environment of the station before Table 2, as the surrounding environment is an important factor affecting the multipath effects, such as: satellite signals being reflected and diffracted by ground obstacles such as buildings.
6. Some sentences contain grammatical mistakes, such as, in line 155 that “it is assume that” would be “it is assumed that”. It is advised to check the entire paper carefully.
7. The format of tables needs more standardization, the units' annotations should be consistent in the paper.
8. Line 371, Figure 8 (b) analyzes the elevation angle of MEO/IGSO satellites to the receiver at different heights with the change of the minimum point angle, while line 371 mentions GEO/IGSO. Please correct this typo.
9. The comparison results of uncorrected code multipath and the correction effects of four methods in Figure 13 are lapped over each other. It is recommended to separate the figure into serval sub-diagrams for the clearer representation of the comparison results..
10. In Figure 9, should the satellite PRN label be added to Figure9 (d)? It is recommended to unified this.
Author Response
Thank you to see the significance of the paper, and we appreciate the friendly suggestions in the manuscript, which are considered useful to further improve the paper. The new changes are highlighted in red color in the revision. For more details, please also see the following replies.

Reviewer 2 Report
Comments and Suggestions for Authors
The results of this work could be implemented for Global Navigation Satellite System receivers and are important for improvement of observation quality and positioning precision.
I have some minor comments and notes:
Lines 183-185: “Ruan conducted detailed modeling of satellite-induced multipath, and proposes that the satellite-induced multipath should be established as functions that are relative to the nadir angle instead of elevation angle [36].”
Comment: In the abstract, line 9, the authors stated that “Multipath effects are the primary unmodeled error sources…” does this statement goes in line with sentence given in lines 183-185?
Lines 235-236: “Multipath errors are usually characterized by spatiotemporal correlation and periodicity.” And lines 238-239: “Moreover, the multipath errors 238 also exhibit periodic repetitive patterns.”
.
Comment: How, and in what studies were revealed these features of multipath? Please give references and some description.
Lines 296-298 : “In order to address this issue, we introduce an attention mechanism that automatically assigns different importance for visible epoch.”
Comment: I will appreciate if the authors describe in some details applied attention mechanism and justify why this mechanism is useful to deal with mentioned issue, and/or give relevant references. P.S. Same comment is relevant for lines 98-116
Lines 332-334: “In our study, we analyzed the BDS dual-frequency observations (B1/B3) from the International GNSS Service (IGS) MGEX station, for 16 days from DOY 160-176, 2021 333 (https://cddis.nasa.gov/)...”
Comment: Why is chosen the time interval equal to 16 days? How does the length of chosen time interval influence results?
Lines: 361-362, Figure 6
Comment: On Figure 6, in normal distribution the second pick is seen. How this second pick could be explained?
Lines 517-520: “Therefore, the proposed AT-Conv-LSTM network could mitigate the multipath efficiently and will be of wide practical value in the fields of standard positioning service and high-precision positioning”
Comment: Is this statement justified by practice? Or implementation of the study results is planned in the future?
Author Response

(The authors gave the same response as above.)

Reviewer 3 Report
Comments and Suggestions for Authors
Multipath effect is an important error that affects the accuracy of global satellite navigation and positioning systems. In the past, it could only be dealt with in some passive ways, but the treatment effect was not very good. The manuscript is based on artificial intelligence algorithms and well solves the impact of multi-path effect on navigation and positioning accuracy. It has very important scientific value and significance and will help further improve the application of global satellite navigation and positioning systems. The content of the manuscript is detailed, the methods are clearly described, and the discussion and analysis are in place. However, there are still some detailed issues, which are listed below. It is recommended that the manuscript be published after minor revision.
L322 and L332: The above two paragraphs can be combined into one paragraph to make the paper structure more reasonable.
L338 and L350: The content of the latter paragraph is too small, so it is not recommended to form a separate paragraph. It can be merged with the previous paragraph.
L406 and L408: The content of the previous paragraph is too little, so it is not recommended to form a separate paragraph. It can be merged with the following paragraph.
Section Discussion: It is recommended to increase the analysis of the limitations of the research content of the manuscript and further work plans in the future.
Author Response
Thank you for your positive comments and valuable suggestions to improve the quality of our manuscript, which are considered useful to further improve the paper. The new changes are highlighted in red color in the revision. For more details, please also see the following replies.

Reviewer 4 Report
Comments and Suggestions for Authors
Please see my comments for the editor.

In general, the English is good, but there are some cases that need to be edited. This becomes critical in a few cases that makes it hard to understand. The text has also some very small paragraphs, like 1.5 lines. They can be merged with other paragraph to make well-sized paragraphs. I highlighted many examples in the PDF file, however, I recommend the text be edited for clearness, punctuations, and grammar.
Author Response
Thank you very much for your careful reading and annotations. Those comments are all valuable and very helpful for revising and improving our paper, as well as the important guiding significance to us. We have studied comments carefully and tried our best to polish the whole paper to improve the language. Thanks for your advice on essay structure and writing habits, I will pay more attention to it in my future essay writing as well. We have listed some significant changes next. For more details, please see the revised manuscript.
